# Exendin-4 Promotes Schwann Cell Survival/Migration and Myelination In Vitro

**DOI:** 10.3390/ijms22062971

**Published:** 2021-03-15

**Authors:** Shizuka Takaku, Masami Tsukamoto, Naoko Niimi, Hideji Yako, Kazunori Sango

**Affiliations:** Diabetic Neuropathy Project, Tokyo Metropolitan Institute of Medical Science, Tokyo 156-8506, Japan; masami1022machida@gmail.com (M.T.); niimi-nk@igakuken.or.jp (N.N.); yako-hd@igakuken.or.jp (H.Y.); sango-kz@igakuken.or.jp (K.S.)

**Keywords:** glucagon-like peptide-1 receptor, exendin-4, IFRS1 Schwann cells, survival, migration, dorsal root ganglion neurons, co-culture, myelination, PI3 kinase signaling pathway

## Abstract

Besides its insulinotropic actions on pancreatic β cells, neuroprotective activities of glucagon-like peptide-1 (GLP-1) have attracted attention. The efficacy of a GLP-1 receptor (GLP-1R) agonist exendin-4 (Ex-4) for functional repair after sciatic nerve injury and amelioration of diabetic peripheral neuropathy (DPN) has been reported; however, the underlying mechanisms remain unclear. In this study, the bioactivities of Ex-4 on immortalized adult rat Schwann cells IFRS1 and adult rat dorsal root ganglion (DRG) neuron–IFRS1 co-culture system were investigated. Localization of GLP-1R in both DRG neurons and IFRS1 cells were confirmed using knockout-validated monoclonal Mab7F38 antibody. Treatment with 100 nM Ex-4 significantly enhanced survival/proliferation and migration of IFRS1 cells, as well as stimulated the movement of IFRS1 cells toward neurites emerging from DRG neuron cell bodies in the co-culture with the upregulation of myelin protein 22 and myelin protein zero. Because Ex-4 induced phosphorylation of serine/threonine-specific protein kinase AKT in these cells and its effects on DRG neurons and IFRS1 cells were attenuated by phosphatidyl inositol-3′-phosphate-kinase (PI3K) inhibitor LY294002, Ex-4 might act on both cells to activate PI3K/AKT signaling pathway, thereby promoting myelination in the co-culture. These findings imply the potential efficacy of Ex-4 toward DPN and other peripheral nerve lesions.

## 1. Introduction

Glucagon-like peptide-1 (GLP-1) is an incretin hormone secreted from intestinal L cells in response to the oral nutrient ingestion and exhibits insulinotropic actions by stimulating specific GLP-1 receptors (GLP-1Rs) on the pancreatic β cells [1]. GLP-1 per se is rapidly degraded by dipeptidyl peptidase type 4 (DPP-4) into the biologically inactive peptide, whereas GLP-1R agonists, such as albiglutide, dulaglutide, exenatide (exendin-4 (Ex-4)), liraglutide, lixisenatide, semaglutide, and taspoglutide, show a substantially longer plasma half-life than GLP-1 because of their resistance to DPP-4. These GLP-1R agonists are currently utilized for the treatment of patients with type 2 diabetes [2]. GLP-1Rs are localized at not only the pancreas but also the extra-pancreatic tissues, including the nervous system [3]. In addition to the inhibition of appetite and food intake, neurotrophic and neuroprotective properties of GLP-1R agonists have been receiving increasing attention. A large number of animal studies have suggested the efficacy of GLP-1R agonists, particularly Ex-4 and liraglutide, for the prevention and restoration of Parkinson’s disease, Alzheimer’s disease, stroke, and other neurodegenerative disorders [4]. According to a recent clinical study [5], Ex-4 ameliorated the severity of motor symptoms associated with Parkinson’s disease. The neuroprotective activities of GLP-1R agonists following axonal injury and in peripheral neuropathies have also been documented [6,7,8]. Himeno et al. [9] reported that Ex-4 restored the decreases in motor nerve conduction velocities and other neurological abnormalities of streptozotocin (STZ)-induced diabetic mice without normalizing blood glucose levels. These findings provide further evidence of the direct actions of Ex-4 on the peripheral nervous system; however, the underlying mechanisms remain largely unclear.

In our previous study [10], Ex-4 dose-dependently (1 nM < 10 nM < 100 nM) enhanced neurite outgrowth and survival of adult rat dorsal root ganglion (DRG) neurons, and these effects were attenuated by co-treatment with phosphatidylinositol-3′-phosphate kinase (PI3K) inhibitor LY294002. In addition, pre-treatment with LY294002 abolished Ex-4-induced suppression of the activity of RhoA, an inhibitory molecule of axonal regeneration and neuronal survival [11]. These findings imply that Ex-4 promotes neurite outgrowth and survival of DRG neurons through the activation of PI3K signaling pathway, which negatively regulates RhoA activity. However, direct evidence that Ex-4 activates PI3K pathway in DRG neurons has not been provided. It also remains unknown if GLP-1R agonists act on Schwann cells to promote axonal regeneration and remyelination after injury or protect neurons against diabetic and other peripheral neuropathies. Liu et al. [6] suggested anti-apoptotic effects of Ex-4 toward Schwann cells in STZ-diabetic rats, but the bioactivities of Ex-4 on cultured Schwann cells have not been fully examined. We have established a spontaneously immortalized Schwann cell line IFRS1 from long-term culture of adult Fischer 344 rat peripheral nerves [12,13]. IFRS1 cells display distinct Schwann cell phenotypes, such as spindle-shaped morphology with intense immunoreactivity for glial cell markers, synthesis and secretion of neurotrophic factors, and fundamental ability to myelinate neurites in co-culture with adult rat DRG neurons [12], NGF-primed PC12 cells [14], NSC-34 mouse neuroblastoma/embryonic spinal motor neuron hybrid cells [15,16] and ND7/23 mouse neuroblastoma/rat embryonic DRG neuron hybrid cells [17] (Takaku et al., in preparation). In addition, we have recently introduced assay systems to evaluate the survival/proliferation and migration of IFRS1 cells [18].

The present study is aimed at elucidating the neuroprotective actions of Ex-4 on cultured rat DRG neurons and IFRS1 Schwann cells. First, we identified GLP-1R on DRG neurons and IFRS1 cells using a well-characterized and verified antibody, then investigated the effects of exogenous Ex-4 on survival/proliferation and migration of IFRS1 cells, and myelination in the DRG neuron–IFRS1 co-culture system.

## 2. Results

### 2.1. GLP-1R mRNA and Protein Expression in Neurons and IFRS1 Schwann Cells

Prior to the evaluation of Ex-4 bioactivities on neurons and Schwann cells, the expression and localization of GLP-1R at these cells should be confirmed. For that purpose, we conducted RT-PCR, Western blot and immunocytochemical analyses. Because much smaller amount of RNA and protein was obtained from primary cultured DRG neurons as compared with the lined cells, it was difficult to perform real-time RT-PCR or quantitative Western blotting using these samples. RT-PCR analysis showed mRNA expression of GLP-1R and glyceraldehyde-3-phosphate dehydrogenase (GAPDH; a housekeeping gene used for confirmation of the proper reactions) in DRG neurons and IFRS1 cells, as well as NSC-34 cells [15] and ND7/23 mouse neuroblastoma/rat embryonic DRG neuron hybrid cells [18] (Figure 1a). These finding imply that these cells synthesize GLP-1R. Western blot analysis and immunofluorescence using knockout-validated antibody Mab7F38 [19] resulted in GLP-1R expression in both DRG neurons and IFRS1 Schwann cells, as well as NSC-34 and ND7/23 cells (Figure 1b,c).

### 2.2. Ex-4 Induces AKT Phosphorylation in ND7/23 and IFRS1 Cells

In our previous study [10], Ex-4 dose-dependently (1 nM < 10 nM < 100 nM) promoted neurite outgrowth and survival of DRG neurons, and these effects were attenuated by co-treatment with a PI3K inhibitor LY294002 (5 μM and 25 μM). We tried to examine phosphorylated state of serine/threonine-specific protein kinase AKT, a key molecule of PI3K signaling pathway in DRG neurons in the presence or absence of Ex-4; however, the amount of protein obtained from the primary cultured DRG neurons was insufficient to conduct Western blotting. We then substituted them with ND7/23 cells, which possess high proliferative activity and retain some characteristic features of DRG neurons [17,18,20]. In addition to ND7/23 cells, the effects of Ex-4 on PI3K signaling in IFRS1 Schwann cells were investigated. Treatment with 100 nM Ex-4 for 60 min significantly upregulated expression of phosphorylated AKT (p-AKT) in both ND7/23 and IFRS1 cells (Figure 2). These findings corroborate our previous study with DRG neurons [10] and suggest the bioactivities of Ex-4 on the neurons and Schwann cells via PI3K/AKT signaling.

### 2.3. Ex-4 Enhances Survival/Proliferation and Migration of IFRS1 Schwann Cells

Because the beneficial effects of Ex-4 on primary cultured and lined DRG neurons have already been documented [9,10,21], the following experiments were conducted with a focus on its bioactivities toward IFRS1 Schwann cells [18]. MTS assays revealed that the values of absorbance at 3 and 6 days of incubation were significantly upregulated by 10 nM and 100 nM Ex-4 (Figure 3a). Scratch wound assays revealed that the number of migrating IFRS1 cells in the 100 nM Ex-4-treated group was significantly higher than that in the control group (Figure 4). However, these Ex-4 effects were attenuated by co-treatment with 25 μM LY294002 (Figure 3b and Figure 4). These findings together with the Ex-4-induced AKT phosphorylation (Figure 2b) suggest that Ex-4 can enhance the survival/proliferation and migration of IFRS1 cells via activating PI3K/AKT signaling pathway.

### 2.4. Ex-4 Stimulates Myelination in DRG Neuron–IFRS1 Co-Culture System

Ex-4 accelerated DRG neuronal cell survival and neurite outgrowth [10] and Schwann cell survival and migration (Figure 3 and Figure 4). These findings together with the previous in vivo study [7] led us to expect Ex-4′s positive effects on myelination in DRG neuron–IFRS1 co-culture system [12,13]. DRG neurons were maintained for a week in a serum-free culture medium with the mixture of neurotrophic factors (10 ng/mL of nerve growth factor (NGF), glial cell line-derived neurotrophic factor (GDNF), and ciliary neurotrophic factor (CNTF)) to promote neurite outgrowth. The neurons were then co-cultured with IFRS1 Schwann cells and maintained for up to 3 weeks under serum-free culture conditions in the presence or absence of 10 nM or 100 nM Ex-4 (Figure 5a). Phase-contrast micrographs at 14 days of co-culture (Figure 5b) showed increased cell-free area among neurite bundles in Ex-4-treated cells compared with Control, suggesting that Ex-4 accelerated the migration of IFRS1 cells toward the neurites emerging from DRG neurons. Immunocytochemistry conducted at 21 days of co-culture indicate that Ex-4 increased the immunoreactivity to peripheral myelin protein (PMP) 22 (green) in IFRS1 cells surrounding βIII tubulin-immunoreactive DRG neurites (red) (Figure 5c). For the quantitative analysis, we counted the number of PMP22-immunoreactive IFRS1 cells attached to a neurite in each photomicrograph, in a similar manner to our previous study [22]. The average number of IFRS1 cells attached to a neurite is 2.4 ± 0.9 in Control and 3.3 ± 0.8 in 100 nM Ex-4 (n ≥ 9 neurites from 3 co-culture samples); the latter is significantly higher than the former (*p* < 0.05) (Figure 5d). Consistent with these findings, Western blot analysis revealed that Ex-4 significantly upregulated the expression of PMP22 and myelin protein zero (MPZ) at 21 days of co-culture (Figure 5e). Furthermore, Ex-4-induced AKT phosphorylation at 2 days of co-culture (Figure 5f) agrees with the involvement of PI3K/AKT signaling pathway in the stimulatory effects of Ex-4 toward both DRG neurons [10] and IFRS1 cells (Figure 2 and Figure 3).

## 3. Discussion

Neuroprotective properties of Ex-4 [6,7,8,9,10] have suggested its therapeutic potential toward DPN independent of the blood glucose-lowering effects [4]. Because neurons and Schwann cells are major constituents of the peripheral nervous system, the present study is focusing on the biological activities of Ex-4 on these cells with possible action mechanisms. Considering that some characteristic features of DRG neurons and Schwann cells change with maturation and aging, these cells obtained from adult animals appear to be better tools than those from embryonic or neonatal animals for the study of DPN and other peripheral nerve lesions [13,23]. The primary culture of DRG neurons from mature rodents have been widely utilized for morphological analyses (for example, in situ hybridization, immunocytochemistry, and electron microscopy) and functional assays (for example, neurite outgrowth, survival, calcium imaging, and electrophysiology). However, the yield of neurons obtained from a single animal is insufficient for conducting quantitative molecular and biochemical analyses (for example, real-time RT-PCR, Western blotting, and enzyme-linked immunosorbent assay), and sacrificing a large number of animals for the quantitative studies is ethically problematic. We then substituted them with ND7/23 cells to perform Western blotting, as mentioned above. The primary culture of adult rodent Schwann cells has been established, but it needs a time-consuming process to get good yields of Schwann cells with high purity from the connective tissue-enriched mature peripheral nerves [23,24]. Although growth stimulants such as forskolin and neuregulin-1β are employed for the passage of IFRS1 cells, they possess proliferative activity feasible for the molecular and biochemical analyses, as well as characteristic features of Schwann cells [12,13,14,16]. In addition to IFRS1 cells in this study, Ex-4 has been shown to enhance proliferation/survival and migration of primary cultured Schwann cells [25] and immortalized mouse Schwann cells 1970C3 [26] (Appendix A). These findings suggest the Ex-4′s stimulatory effects on Schwann cells regardless of cell types (primary or lines) and lines. Numerous Schwann cell lines have been established by us and others, but only a few of them, including IFRS1 cells, have shown the capability to form myelin structure under co-culture with neuronal cells [13,27]. In particular, our DRG neuron–IFRS1 co-culture system can be stably and effectively used for exploring neuron–Schwann cell interplay during axonal degeneration and regeneration, as well as pathogenesis of demyelinating neuropathies [22].

The existence of GLP-1R in DRG neurons and IFRS1 cells is the prerequisite for investigating the biological activities of Ex-4 on these cells. Although the previous studies indicated GLP-1R immunoreactivity in DRG neurons and Schwann cells in vivo and in vitro [9,10], the specificity of the GLP-1R antibodies employed in those studies has not been verified. Our RT-PCR analysis resulted in GLP-1R mRNA expression in primary cultured DRG neurons and IFRS1 cells, as well as NSC-34 and ND7/23 neuronal cell lines (Figure 1a). The Western blot analysis using knockout-validated monoclonal Mab7F38 antibody [18] confirmed localization of GLP-1R at these cells (Figure 1b). The immunofluorescent micrographs (Figure 1c) indicate ubiquitous distribution of GLP-1R at both cytoplasm and cell membrane rather than cell surface-predominant expression in both DRG neurons and IFRS1 cells. Similarly, GLP-1R immunoreactivity throughout the cytoplasm of murine pancreatic acinar cells was detected by the same antibody [19].

Ex-4-induced AKT phosphorylation in both ND7/23 and IFRS1 cells (Figure 2) and the attenuation of its stimulatory effects on DRG neurons [10] and IFRS1 cells by co-treatment with a PI3K inhibitor LY294002 (Figure 3 and Figure 4) suggest the involvement of PI3K/AKT signaling pathway in its bioactivities toward these cells. Consistent with these findings, GLP-1R agonists alleviated neuronal cell death after subarachnoid hemorrhage in rats [28] and methylglyoxal- and β-amyloid-induced neurotoxicity [29,30] via GLP-1R/PI3K/AKT axis. In the previous studies using DRG neurons [10] and SH-SY5Y and PC12 cells [31], neurotrophic and neuroprotective actions of Ex-4 can be, at least partially, attributable to the suppression of RhoA, an inhibitory molecule of axonal regeneration [32], through PI3K/AKT pathway. The downstream target molecules of PI3K pathway, such as GSK3β and mTOR, may also be related to the neuroprotective function of GLP-1R agonists [33,34]. In addition to those studies targeted at neurons, restoring effects of Ex-4 against decreased myelinated nerve fiber size and Schwann cell apoptosis in STZ-diabetic rats have been documented [6]; however, direct actions of Ex-4 and other GLP-1R agonists toward Schwann cells with possible mechanisms remain unclear. Like Ex-4 in this study, several molecules have been shown to promote Schwann cell viability/proliferation and migration through PI3K/AKT pathway [35,36,37]. Thus, this pathway appears to play a key role in the neuroprotective properties of Schwann cells in response to peripheral nerve lesions. In a recent study by Pan et al. [25], Ex-4 enhanced Schwann cell proliferation and migration through activating JAK/STAT pathway in vitro. Although we have not examined whether JAK/STAT pathway is involved in the Ex-4′s stimulatory effects on IFRS1 cells, Ex-4 and other GLP-1R agonists are likely to exert their neuroprotective actions via activating multiple signaling pathways. For instance, stimulatory effects of Ex-4 on rat colonic myenteric neurons are dependent on activation of PI3K and mitogen-activated protein kinase (MEK/ERK) signaling cascades [38]. Our future studies using IFRS1 cells will focus on the interrelationship between PI3K/AKT, MEK/ERK, and JAK/STAT pathways regarding the neuroprotective function of Ex-4. Which pathways are more involved in Schwann cell survival/proliferation and/or migration may also be an important issue to be solved. Our previous study using DRG neurons suggest the involvement of the above 3 signaling pathways in CNTF–induced neurite outgrowth, whereas PI3K/AKT and JAK/STAT pathways are thought to play major roles in mediating the survival response of neurons to CNTF [39].

The findings from the co-culture system (Figure 5) imply that Ex-4 can accelerate the myelination process, and support the previous studies addressing Ex-4-induced axonal regeneration and remyelination in rats following sciatic nerve crush injury [7]. Although our trials to detect nodal structures in the co-culture have not been successful, we had verified the formation of myelin sheaths by Sudan black B staining and electron microscopy [12,13]. In addition, double immunofluorescence (MPZ or PMP22/ βIII tubulin) combined with X-Gal staining on the co-cultures of DRG neurons and β-galactosidase gene-inserted IFRS1 cells revealed co-localization of the X-Gal stain with the myelin protein expression. By electron microscopy, the X-Gal stained IFRS1 cells showed cytoplasm with granular electron-dense precipitate and a myelin sheath [12]. Based on these findings, it seems appropriate to provide quantitative data of myelin sheath formation using double immunofluorescence (Figure 5c,d) [22]. The morphological evidence is further supported by Ex-4-induced upregulation of PMP22 and MPZ protein expression (Figure 5e). Because the beneficial effects of Ex-4 on DRG neurons and IFRS1 cells were abolished by the PI3K inhibitor, Ex-4 might act on both cells to accelerate myelination through PI3K/AKT signaling pathway. In addition, Ex-4-induced AKT phosphorylation at 2 days of co-culture (Figure 5f) agrees with the previous study, which suggested that intracellular signals mediated by PI3K/AKT are crucial for initiation of myelination [40,41]. However, there is room for further investigation to safely state that AKT signaling pathway activated by Ex-4 in both cells play a major role in myelination in the co-culture system. In addition to the analyses using the PI3K inhibitor, we plan to manipulate GLP-1R and AKT genes in either DRG neurons or IFRS1 Schwann cells. Because impaired axonal regeneration and remyelination are one of the characteristic features in pathophysiology of DPN [42], therapeutic approaches of Ex-4 toward DPN can be expected. Sheknova et al. [43] recently reported that treatment of STZ-diabetic rats with arginine-rich Ex-4 ameliorated neuropathic pain, as well as reduced myelinated nerve fiber diameters and myelin basic protein expression in sciatic nerves. Our in vitro studies combined with that in vivo study suggest the efficacy of Ex-4 for ameliorating the loss of myelinated fibers in DPN. Using the same co-culture system, we have observed myelination-promoting activities of GDNF [12] and CNTF [13]. The efficacy of these neurotrophic factors for axonal regeneration after injury and amelioration of DPN have been shown in animal models [44]; however, the clinical trials using them thus far revealed unsuccessful or incomplete [45]. In contrast, Ex-4 has been approved as a safe and effective anti-diabetic agent [2] and its repurposing for patients with Parkinson’s diseases is an ongoing project [5,46]. Although the clinical trials of Ex-4 and other GLP-1 mimetics targeted at patients with DPN have not gained significant outcomes [47,48], elucidating their precise action mechanisms will help advance the therapeutic strategies.

The beneficial effects of Ex-4 on DRG neurons, IFRS1 Schwann cells, and their co-culture system in our previous [10] and present studies are schematically represented in Figure 6. Although further analyses are needed to strengthen our hypothesis, these findings imply their efficacy for the prevention and restoration of DPN and other PNS lesions.

## 4. Materials and Methods

### 4.1. Animals

Six-week-old female Wistar rats were obtained from CLEA Japan, Inc. (Shizuoka, Japan). All rats received humane care and handling in accordance with the Guidelines of the Care and Use of Animals (Tokyo Metropolitan Institute of Medical Science, 2011; institutional approval number 20-016). Prior to the dissection, rats were anesthetized for euthanasia with 3% isofluorane (Abbott Japan, Tokyo, Japan) for 3 min.

### 4.2. Isolation and Culture of DRG Neurons

Dissociated cell culture of DRG neurons was performed as previously reported method [10,18]. Briefly, DRG from the cervical to the lumbar level were dissected from each animal and dissociated with collagenase (CLS-3; Worthington Biochemicals, Freehold, NJ, USA) and trypsin (Sigma, St. Louis, MO, USA). These ganglia were subjected to density gradient centrifugation (5 min, 200 g) with 30% Percoll PLUS^TM^ (GE Healthcare Bio-Sciences Corp., Piscataway, NJ, USA) to eliminate the myelin sheath. This procedure resulted in a yield of more than 5 × 10^4^ neurons along with a smaller number of non-neuronal cells. These neurons were suspended in Dulbecco’s Modified Eagle’s medium (DMEM)/Ham’s F12 (Thermo Fisher Scientific Inc., Waltham, MA, USA) supplemented with 10% fetal bovine serum (FBS; Thermo Fisher Scientific Inc., Waltham, MA, USA) and employed for Western blot and immunocytochemical analyses and co-culture with IFRS1 Schwann cells.

### 4.3. Culture of IFRS1 Schwann Cells, NSC-34 Motor Neuron-Like Cells, and ND7/23 Sensory Neuron-Like Cells

IFRS1 Schwann cells were established in our laboratory [12]. IFRS1 cells at the passage of 30-40 were maintained in DMEM (Thermo Fisher Scientific Inc., Waltham, MA, USA) supplemented with 5% FBS; and employed for Western blot and immunocytochemical analyses, proliferation/survival, and migration assays; and co-culture with DRG neurons. NSC-34 cells [15] and ND7/23 cells [17] were kindly provided by Kazuhiko Watabe of Kyorin University, and Atsufumi Kawabata of Kindai University, respectively. These cells at the passage of 15–20 were maintained in DMEM supplemented with 5% FBS and employed for Western blot analysis.

### 4.4. Proliferation/Survival Assay for IFRS1 Cells

The effects of Ex-4 on the proliferation/survival of IFRS1 cells were evaluated using the CellTiter 96^®^ Aqueous One Solution Cell Proliferation Assay kit (Promega Corp., Madison, WI, USA), as previously reported [18]. The cells were seeded onto each well of 96-well culture plates at an approximate density of 3 × 10^4^/cm^2^ and were incubated overnight in DMEM supplemented with 5% FBS. The cells were then maintained in DMEM/1% FBS with different concentrations (0, 10, or 100 μM) of Ex-4 (R&D Systems, Inc., Minneapolis, MN, USA) for 1, 3, and 6 days. After rinsing with 250 μL of DMEM, the cells were incubated for 1 h at 37 °C in 100 μL of DMEM with 10 μL of CellTiter 96^®^ Aqueous One Solution Reagent, and absorbance at 490 nm was determined with a microplate reader (Varioskan Flash; Thermo Fischer Scientific Inc., Waltham, MA, USA).

### 4.5. Migration Assay for IFRS1 Cells

The effects of Ex-4 on IFRS1 cell migration were evaluated by the scratch wound assay, as previously reported [18,49]. Briefly, the cells were seeded onto a poly-L-lysine (PL; Sigma, St. Louis, MO, USA, 10 μg/mL)-coated 35 mm glass-bottomed dish with grid (Matsunami Glass Ind., LTD, Osaka, Japan) at an approximate density of 2 × 10^4^/cm^2^ and maintained in DMEM/5%FBS for 24 h. Then, a cell-free area was scratched using a sterile 200-microliter pipette tip (BM Equipment Co., Ltd., Tokyo, Japan). The cells were incubated in DMEM/1%FBS with different concentrations (0, 10, or 100 μM) of Ex-4, and the scratch was photographed 2 and 26 h after its generation using a phase-contrast light microscope (IMT-2; Olympus, Tokyo, Japan) equipped with a microscope digital camera system (DP22-CU; Olympus, Tokyo, Japan) and image analysis software (WinROOF2015; Mitani Corporation, Tokyo, Japan). The number of cells migrating into the square area of scratch (6 × 6 grids) during 24 h was calculated by reducing the number in the area at 26 h from that at 2 h.

### 4.6. In Vitro Myelination

Myelinating co-culture of DRG neurons and IFRS1 cells was performed as previously reported [12] with slight modifications. Briefly, DRG neurons were seeded on type I collagen-coated 2-well chamber slides (Matsunami Glass Ind., LTD, Osaka, Japan) and Aclar fluorocarbon coverslips (Nissin EM Co., Tokyo, Japan) at an approximate density of 2 × 10^3^/cm^2^ and maintained for 5-7 days in DMEM/F12 containing N2 supplement (Thermo Fisher Scientific Inc., Waltham, MA, USA), 10 ng/mL nerve growth factor (NGF; R&D Systems, Inc., Minneapolis, MN, USA), 10 ng/mL glial cell line-derived neurotrophic factor (GDNF; R&D Systems, Inc., Minneapolis, MN, USA), and 10 ng/mL ciliary neurotrophic factor (CNTF; PeproTech, Rocky Hill, NJ, USA). The cell density ratio of DRG neurons to IFRS1 cells was adjusted to 1:10. The co-cultured DRG neurons and IFRS1 cells were incubated for 2 days in DMEM/F12 supplemented with 5% FBS and then maintained for up to 28 days in DMEM/F12 containing B27 supplement (Thermo Fisher Scientific Inc., Waltham, MA, USA), 50 μg/mL ascorbic acid (FUJIFILM Wako Pure Chemical Corp., Osaka, Japan), and different concentrations (0, 10, or 100 μM) of Ex-4.

### 4.7. Reverse Transcription-Polymerase Chain Reaction (RT-PCR) for the Detection of GLP-1R mRNA

Total RNA was isolated from the cultured cells using a RNeasy Mini Kit (Qiagen K.K., Tokyo, Japan). RT-PCR analysis was performed using the methods described in a previous paper [12], and the GoTaq Green Mater Mix (Promega Corp., Madison, WI, USA) and MiniOpticon (Bio-Rad Laboratories, Inc., Tokyo, Japan) were used according to the manufacturer’s instructions. PCR was performed for 25–33 cycles at 95 °C for 30 s, 55 or 60 °C for 30 s, and 72 °C for 30 s. The sequences of oligonucleotide sense and antisense primers of GLP-1R and GAPDH (a house-keeping gene) for PCR are as follows:

GLP-1R, sense primer, 5′- TATCTTCATCCGTGTCATCTGC -3′ and antisense primer, 5′- AGCAGTACAGGATAGCCACCAT -3′ (a 245-bp product);

GAPDH, sense primer, 5′-TTCAACGGCACAGTCAAGGCTG-3′ and antisense primer, 5′-TGGCATGGACTGTGGTCATGAG-3′ (a 376-bp product).

The PCR products were separated by electrophoresis using a 2% agarose gel and visualized with ethidium bromide staining.

### 4.8. Western Blot Analysis

Western blot analysis was performed as previously described [18] with slight modifications. Briefly, protein was extracted from the cultured cells using 1× sodium dodecyl sulfate (SDS) sample buffer. The cell extracts were resolved using SDS-polyacrylamide gel electrophoresis (SDS-PAGE) in 5–20% SDS-PAGE gel (FUJIFILM Wako Pure Chemical Corp., Osaka, Japan) and transferred onto a polyvinylidene fluoride membrane using an electroblotter (Nihon Eido Co., Ltd., Tokyo, Japan). The membrane was incubated in phosphate-buffered saline (PBS) with 0.1% Tween 20 (including 5% skimmed milk or 3% bovine serum albumin) for 1 h at room temperature and then overnight at 4 °C with the following antibodies:mouse anti-GLP-1R monoclonal antibody (1:1,000; Mab7F38, Developmental Studies Hybridoma Bank, Iowa City, USA) [19],rabbit anti-AKT polyclonal antibody (1:1,000; Cell Signaling Technology, Beverly, MA, USA),rabbit anti-phospho-AKT polyclonal antibody (1:1,000; Cell Signaling Technology, Beverly, MA, USA),rabbit anti-myelin protein zero (MPZ) polyclonal antibody (1:1,000; a kind gift from Dr. Hitoshi Nagai, Mitsubishi Tanabe Pharma Corporation, Yokohama, Japan) [14],rabbit anti-peripheral myelin protein 22 (PMP22) polyclonal antibody (1:1,000, Sigma, St. Louis, MO, USA) [50], andmouse anti-β-tubulin isotype I+II monoclonal antibody (1:3,000; Sigma, St. Louis, MO, USA).

After rinsing with PBS, the membrane was incubated in a solution of horse radish peroxidase-conjugated anti-rabbit IgG antibody or anti-mouse IgG antibody (1:2000; MBL Corp., Ltd., Nagoya, Japan) for 1 h. After rinsing, immunocomplexes on the membrane were visualized with ECL plus a Western Lighting Ultra (PerkinElmer, Inc., Waltham, MA, USA). Ez-Capture II chemiluminescence imaging system (Atto Corp, Tokyo, Japan) was used for quantitative analysis, and the relative signal intensity of PMP22/MPZ and phosphor-AKT was normalized to the intensity of β-tubulin I+II (Figure 5d) and AKT (Figure 5e), respectively.

### 4.9. Immunofluorescence

DRG neurons, IFRS1 cells, and their co-cultures were fixed with 4% paraformaldehyde for 10 min at 4 °C and then treated with 0.1% Triton X-100 in PBS for 5 min at room temperature. The cells were incubated overnight at 4 °C with the following antibodies diluted with 20 mM PBS containing 0.4% Block Ace (DS Pharma Biomedical Co., Osaka, Japan):mouse anti-GLP-1R monoclonal antibody (1:1,000) [19],rabbit anti-PMP22 polyclonal antibody (1:1,000) [50], andmouse anti-βIII tubulin monoclonal antibody (1:1,000; Sigma, St. Louis, MO, USA) [51].

After rinsing with PBS, the cells were incubated for 1 h at 37 °C with Alexa Fluor 488 anti-rabbit IgG antibody (1:200, Thermo Fisher) and/or Alexa Fluor 594 or 488 anti-mouse IgG antibody (1:200, Thermo Fisher) and then incubated with 4’,6-diamidino-2-phenylindole (DAPI; 300 nM, Thermo Fisher) for 5 min at room temperature. The samples processed for immunofluorescence were observed and recorded using a TCS SP5 confocal microscope system (Leica Microsystems, Wetzlar, Germany). Immunocytochemical controls in which the primary antibodies were omitted resulted in lack of positive staining on each culture sample (data not shown). Characterization and validation of these antibodies were described in the above articles.

### 4.10. Statistical Analysis

All data were presented as means with standard deviation. The number of experiments is indicated in the figure legends. Parametric comparisons between experimental groups were performed by one-way analysis of variance (ANOVA); when ANOVA showed a significant difference between groups (*p* < 0.05), Tukey-Kramer test was used to identify which group differences accounted for the significant *p* value.

## Figures and Tables

**Figure 1 ijms-22-02971-f001:**
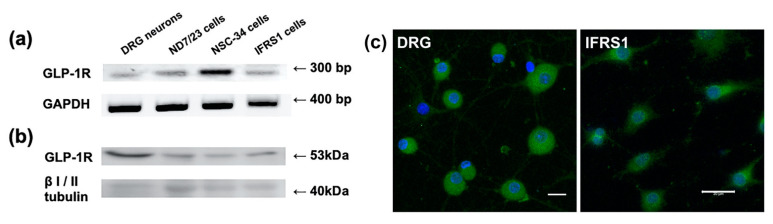
Glucagon-like peptide-1 (GLP-1R) mRNA and protein expression in primary cultured dorsal root ganglion (DRG) neurons, NSC-34 and ND7/23 neuronal cell lines, and an IFRS1 Schwann cell line. (**a**) GLP-1R and glyceraldehyde-3-phosphate dehydrogenase (GAPDH) mRNA expression in the 4 kinds of cells; RT-PCR analysis. (**b**) GLP-1R and βI/II tubulin (a house-keeping protein used for confirmation of the proper reactions) expression in the 4 kinds of cells; Western blot analysis. (**c**) GLP-1R immunoreactivity in DRG neurons and IFRS1 cells; immunofluorescence. Nuclei are stained blue with 4’,6-diamidino-2-phenylindole (DAPI). Scale bars indicate 20 μm.

**Figure 2 ijms-22-02971-f002:**
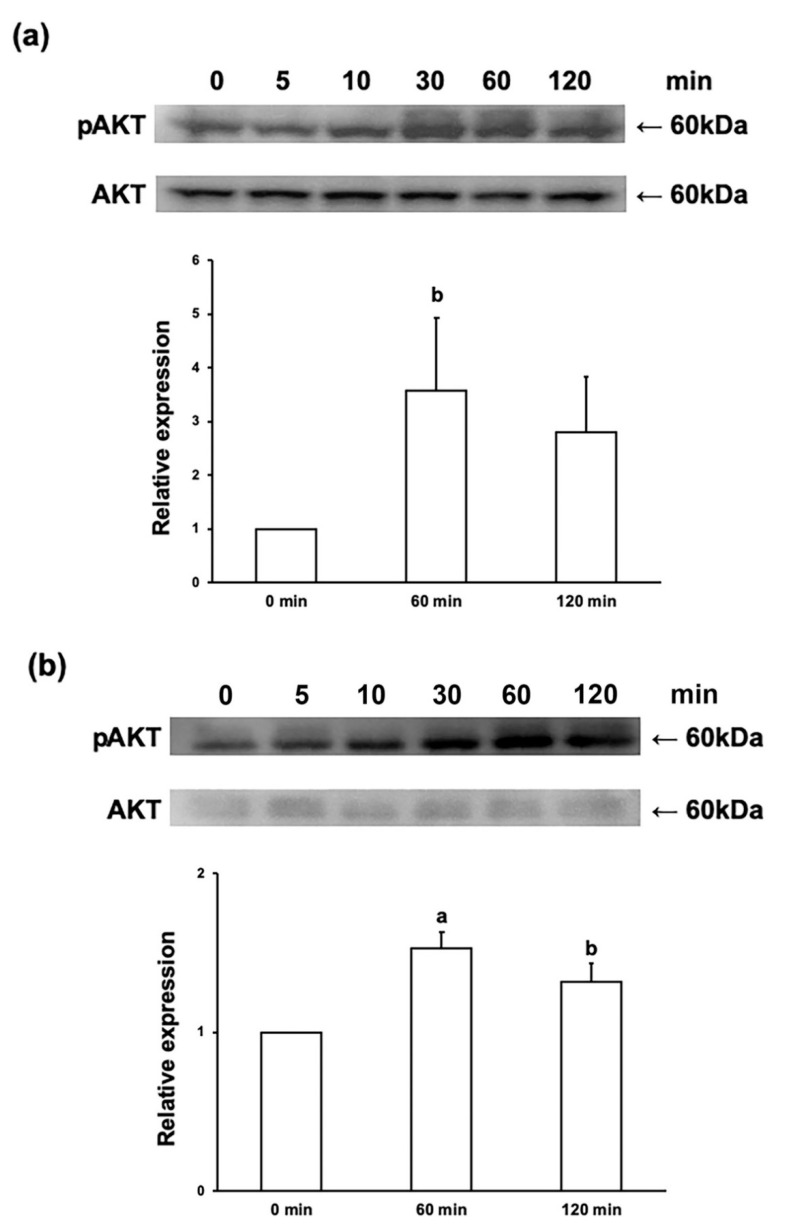
Treatment with 100 nM exendin-4 (Ex-4) induces phosphorylation of serine/threonine-specific protein kinase AKT in ND7/23 cells (**a**) and IFRS1 cells (**b**). The representative pictures of the Western blot analysis (upper) and quantitative data (relative phosphorylated AKT (p-AKT) expression at 0, 60, and 120 min after Ex-4 treatment; lower) are shown. Values represent means + SD from 3 experiments. a: *p* < 0.01 as compared with 0 min, b: *p* < 0.05 as compared with 0 min.

**Figure 3 ijms-22-02971-f003:**
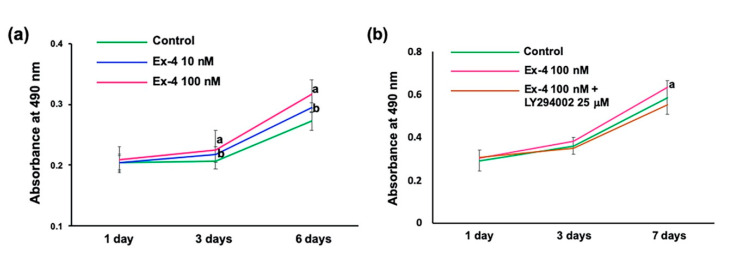
Ex-4 promotes survival/proliferation of IFRS1 cells; MTS assay. (**a**) The absorbance at 1, 3, and 6 days after treatment with 0 (Control), 10 and 100 nM Ex-4. Values represent means ± SD from 18 experiments. a: *p* < 0.01 as compared with Control, b: *p* < 0.05 as compared with Control. (**b**) The absorbance at 1, 3, and 7 days after treatment with 0 (Control) and 100 nM Ex-4 in the presence or absence of 25 μM LY294002. Values represent means ± SD from 18 experiments. a: *p* < 0.01 as compared with Control and Ex-4 100 nM + LY294002 25 μM.

**Figure 4 ijms-22-02971-f004:**
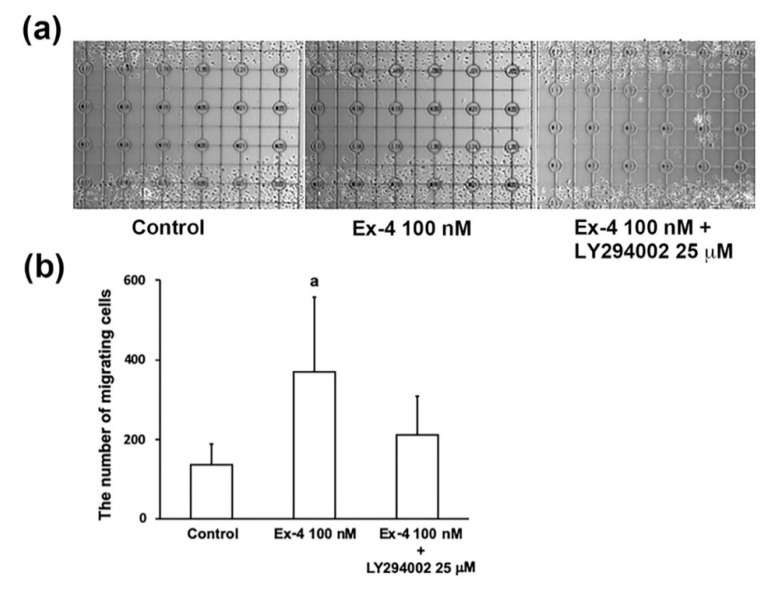
Ex-4 promotes migration of IFRS1 cells; scratch wound assay. (**a**) Representative photomicrographs of IFRS1 cells at 1 day after scratch. (**b**) The number of migrating cells at 1 day after treatment with 0 (Control) and 100 nM Ex-4 in the presence or absence of 25 μM LY294002. Values represent means + SD from 24 experiments. a: *p* < 0.01 as compared with Control and Ex-4 100 nM + LY294002 25 μM.

**Figure 5 ijms-22-02971-f005:**
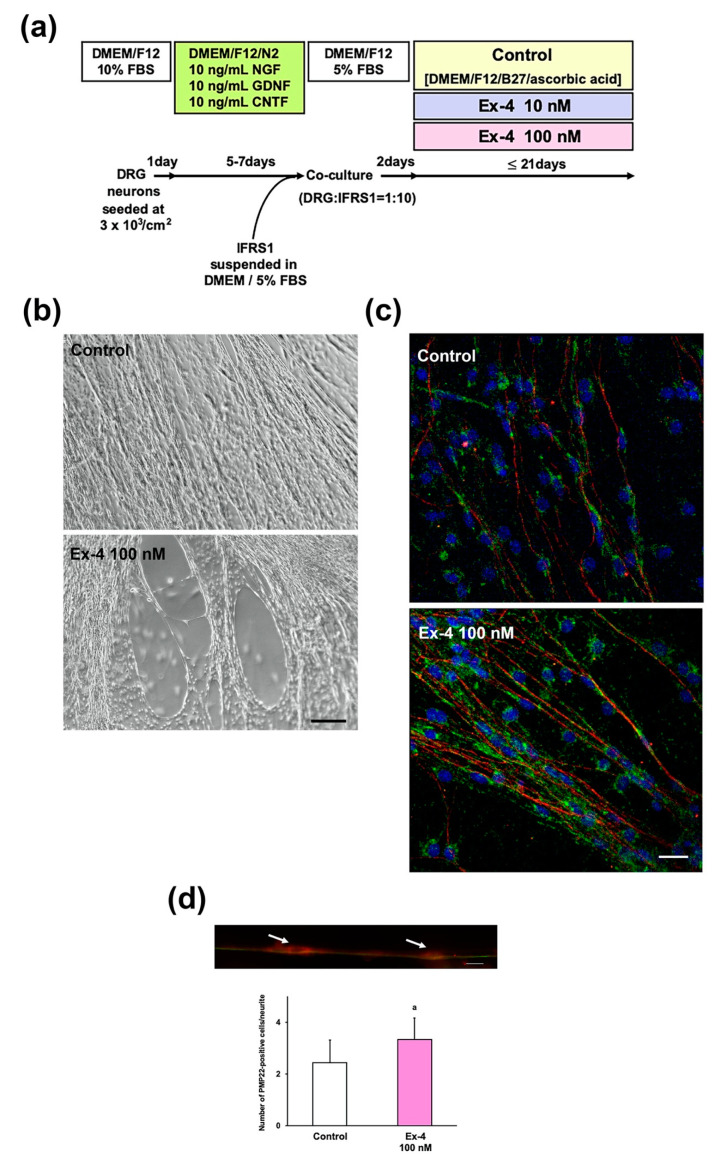
Ex-4 stimulates myelination in DRG neuron–IFRS1 co-culture system. (**a**) Schematic representation of the standard procedure for the co-culture of adult rat DRG neurons and IFRS1 cells. (**b**) Representative phase-contrast photomicrographs at 14 days of co-culture in the presence or absence of 100 nM Ex-4. Scale bar indicates 100 μm. (**c**) Representative immunofluorescence photomicrographs at 21 days of co-culture in the presence or absence of 100 nM Ex-4. Ex-4 increased the immunoreactivity to rabbit anti-peripheral myelin protein 22 (PMP22) (green) in IFRS1 cells surrounding βIII tubulin-immunoreactive DRG neurites (red). Scale bar indicates 20 μm. (**d**) A representative immunofluorescence photomicrograph at 21 days of co-culture (upper) shows two PMP-immunoreactive cells (arrows) attached to a βIII tubulin-immunoreactive neurite. Scale bar indicates 10 μm. Ex-4 increases PMP22-immunoreactive IFRS1 cells attached to a neurite (lower). Values represent means + SD from 9 to 16 experiments. a: *p* < 0.05 as compared with Control. (**e**) Ex-4 upregulates expression of myelin proteins PMP22 and MPZ at 21 days of co-culture; Western blot analysis. Values represent means + SD from 3 to 6 experiments. a: *p* < 0.05 as compared with Control. (**f**) Ex-4 induces phosphorylation of AKT at 2 days of co-culture; Western blot analysis. Values (relative p-AKT expression) represent means + SD from 6 experiments. a: *p* < 0.05 as compared with Control.

**Figure 6 ijms-22-02971-f006:**
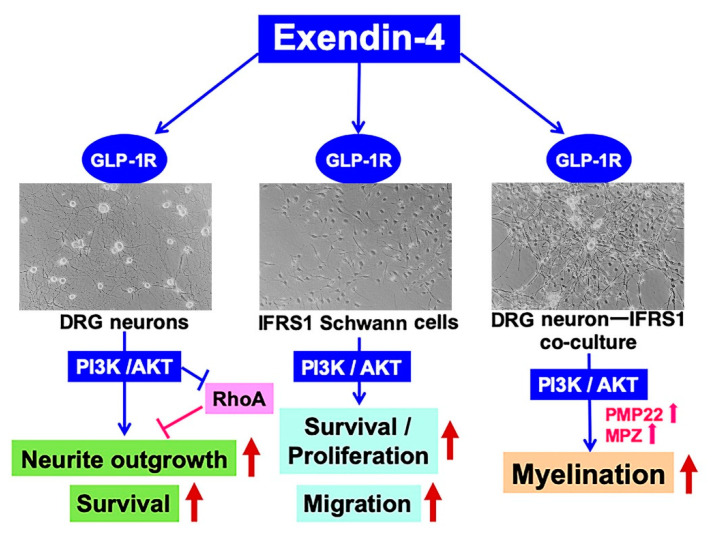
Schematic representation of possible mechanisms for the beneficial effects of Ex-4 on DRG neurons, IFRS1 Schwann cells, and their co-culture system.

## Data Availability

The data presented in this study are available on request from the corresponding author.

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
