# Peer review of "Exendin-4 Promotes Schwann Cell Survival/Migration and Myelination In Vitro"

_ijms, 2021, doi:10.3390/ijms22062971_

Round 1

Reviewer 1 Report

The authors show the promotion effect of Ex-4 on cultured DRG neurons and IFRS1 Schwann cells in proliferation and myelination. As a mechanism, they suggest GLP-1R on neurons and IFRS1 cells.

Overall, the authors need to improve the quality and representation of figures.
For example, in Fig 5. (d). the PMP22 intensity of two bands treated EX-4 10nM looks very different, but the authors took them statistics to create a graph. It may contain a significant error.

Figure 5 (b) and (C) show that cells grow with alignment by EX-4 treatment, in contrast to control. Is this the effect of Ex-4? Or is it a treatment effect?

The Authors used several cell lines for neurons but only used one line of IFRS1 for Schwann cell, which is too limited to show the proliferation/migration or myelination of the Schwann cell. What about other kinds of cell lines or primary cultures as Schwann cell? It would be better to show results from additional cell lines or primary cultures.

Author Response

Thank you for advice. Please see the attachment.

Reviewer 2 Report

The manuscript by Shizuka Takaku end colleagues addresses the effect of stimulation of GLP-1 receptors on immortalized adult rat Schwann cells (line IFRS1) and adult rat DRG neuron–IFRS1 co-culture system by use of their agonist exendin-4.

First, localization of GLP-1R in both DRG neurons and IFRS1 cells was confirmed. The authors then concluded that treatment with Ex-4 significantly improved proliferation and migration of IFRS1 cells. Then they claimed that the movement of IFRS1 cells toward neurites emerging from DRG neuron cell bodies in the co-culture and axons myelination was accelerated. In support of the latter, myelin 22 protein and myelin zero protein have been reported to be upregulated in co-cultures. Finally, they concluded that Ex-4 could activate GLP-1R on both cell types, which resulted in the activation of the PI3K / AKT signaling and myelination pathway in co-culture.

The study focuses on interesting observations and possibly a new mechanism and role for GLP-1R activation in myelination. However, there are some important points to consider.

 Specific comments:

  1. Figure 1 compares GLP-1R mRNA and protein levels in several cell types. A quantitative analysis should be performed here and the inconsistency between gene expression and protein level should be discussed, if it is still observed in several replicates. Now it seems that the mRNA level in DRGs is significantly lower than in other cell types, while the protein level appears to be highest in the same DRG cells. Is it consistent across replications? It seems to be misleading without any explanation.
  2. Similarly, data presented in fig.2 should be quantified.
  3. The most important issue is related to presentation of results coming from co-culture myelination experiments. The authors claim the cultures myelinated but they present only upregulation of PLP22 and P0 protein level. This is insufficient to conclude that Ex-4 enhances myelin sheaths formation. Presenting Western blot data does not, of itself, provide a formal functional link with the physiological relevance. It needs to be clarified whether the Schwann cells, which upregulate myelin proteins, eventually produce functional myelin sheaths. In this case, it would be beneficial to characterize the nodal structures in myelinated co-cultures using antibodies directed against proteins located in the specialized axon domains that form in the Ranvier node or paranodes, and to quantify the myelin segments.
  4. What data might support the conclusion that the movement of IFRS1 cells toward DRG neurites in the co-culture was accelerated? This needs to be precisely clarified.
  5. Finally, a question arises regarding the proposed molecular mechanism: can AKT activation be considered as a regulatory mechanism for Ex-4 driven myelination by acting on both cell types simultaneously? The data presented here do not support the thesis that activation of AKT by Ex-4 in both cell types is necessary to enhance myelination. The conclusion that Ex-4 acts on the both cell types to accelerate myelination because the effect of was abolished by the AKT inhibitor in co-culture is invalid. To resolve this question, an analysis of myelination in co-cultures with GLP-1R genetic knockout induced specifically in either DRG or Schwann cells would be helpful.

Author Response

(The authors gave the same response as above.)

Round 2

Reviewer 2 Report

The manuscript has been significantly improved to the reviewer's advice.